# VISCOS Flows: Variational Schur Conditional Sampling with Normalizing Flows

## Abstract

We present a method for conditional sampling for (possibly pre-trained) normalizing flows when only part of an observation is available. We derive a lower bound to the conditioning variable log-probability using Schur complement properties in the spirit of Gaussian conditional sampling. Our derivation relies on partitioning flow's domain in such a way that the flow restrictions to subdomains remain bijective, which is crucial for the Schur complement application. Simulation from the variational conditional flow then amends to solving an equality constraint. Our contribution is three-fold: a) we provide detailed insights on the choice of variational distributions; b) we discuss how to partition the input space of the flow to preserve bijectivity property; c) we propose a set of methods to optimise the variational distribution. Our numerical results indicate that our sampling method can be successfully applied to invertible residual networks for inference and classification.

## 1 Introduction

Conditional data generation is a ubiquitous and challenging problem, even more so when the data is high dimensional. If the partitioning of the conditioned data and conditioning itself can be established in advance, a wide variety of inference tools can be used ranging from Bayesian inference (Gelman, 2013) and its approximations (e.g. variational inference (Klys et al., 2018)) to Gaussian (Williams & Rasmussen, 2006) or Neural Processes (Garnelo et al., 2018). The task is considerably more challenging when the partitioning cannot be anticipated. This is met in a wide range of applications, from few shots learning (Wang et al., 2020) to super resolution image generation (Bashir et al., 2021), high-dimensional Bayesian optimisation (Moriconi et al., 2020; Gomez-Bombarelli et al., 2018), learning with incomplete datasets (Yoon et al., 2018; Li et al., 2019; Richardson et al., 2020), image inpainting (Elharrouss et al., 2020) and many more.

If we can assume that the data is normally distributed, the conditional distribution can be computed in closed form using Gaussian elimination and *the Schur complement*. This observation paved the way for Gaussian Processes (GPs) solutions (Williams & Rasmussen, 2006) and alike. Although they constitute a solid component of the modern machine learning toolbox, the Gaussian assumption is quite restrictive in nature. Indeed: GPs in their vanilla form are used when the observations are uni-dimensional and when inference has to be performed on a subset of these variables. As a consequence, alternative tools may need to be used when dealing with even moderately dimensional data, or when part of the input data can be missing.

It is well-established that normalizing flows (NFs) (Rezende & Mohamed, 2015) can model arbitrarily complex joint distributions. Furthermore, they naturally embed together the joint probability of any partition of data that could occur during training or at test time. Hence, it is tempting to make use of NFs to perform conditional sampling, but only a few works have provided solutions each with some limitations. (Cannella et al., 2020) proposed Projected Latent Monte Carlo Markov Chains (PL-MCMC) to generate samples whose probability is guaranteed to converge to the desired conditional distribution as the chain gets longer. While such convergence is certainly an advantage, one can identify several important drawbacks: first, as for every other Monte Carlo sampling method, the mixing time cannot be known in advance, and hence one may wonder whether the samples that are gathered truly belong to the conditional distribution. Second, although the observed part of the imputed data converges towards the true values, a difference between the two may persist. Third,

training a model comprising of missing data with PL-MCMC is achieved through a Monte Carlo Expectation Maximisation (MCEM) scheme (Dempster et al., 1977; Neath, 2013). Under appropriate assumptions, MCEM is guaranteed to converge to a local maximiser of the log-likelihood function, but this is conditioned on the quality of the data generated by the Monte Carlo algorithm. Consequently, as the optimisation progresses towards the optimum, longer chains may be required to ensure convergence or even to obtain convincing results (Neath, 2013). MCFlow (Richardson et al., 2020) relies on an auxiliary feedforward neural network whose role is to produce latent embeddings with maximum a posteriori likelihood values. Those values are constrained to lie on the manifold of latent vectors whose mapping to the observed space match the observations. MCFlow produces state-of-the-art results in terms of image quality or classification accuracy when compared to GAN-based methods such as (Yoon et al., 2018; Li et al., 2019). However, this method requires a set of adjustments to the model and needs retraining for any additional incomplete data. On the other hand, ACFlow Li et al. (2020) learns all conditional distributions for all possible masking operations, which can be quite computationally expensive. As such, both MCFlow and ACFlow cannot be applied to post-training data completion as PL-MCMC.

Several conditioning techniques have been used with NFs in contexts where the joint probability distribution of the conditioning and conditioned random variable is of no interest. For instance, (Rezende & Mohamed, 2015; van den Berg et al., 2018) extended the amortisation technique used in Variational Auto-Encoders (VAEs) (Kingma & Welling, 2013) to the parameters of the flow. (Winkler et al., 2019) extended this use to the scenario of conditional likelihood estimation, while (Trippe & Turner, 2018) provided a Variational Bayes view on this problem by using a Bayesian Neural Network for conditioning feature embedding. On a different line of work, (Kingma & Dhariwal, 2018) use a heuristic form of *a posteriori* conditioning. The generative flow is trained with a classifier over the latent space, which forces the latent representation location to be indicative of the class they belong to. At test time, some parametric distribution is fitted on the latent representations to account for the attributes of the images that are to be generated. (Nguyen et al., 2019) used a hybrid approach, whereby the conditioning and conditioned variables joint distribution is modelled by a normalizing flow in such a way that conditional sampling of one given the other is straightforward to apply. Still, either with this approach or the others above, the knowledge of what part of the data constitutes the conditioning random variable is required beforehand.

**Our Contribution**: We propose a conditional NF sampling method in the following setting: a) conditional data generation is performed after a model has been trained *without* taking conditional sampling into account; b) training data may be itself incomplete, in the sense that some training features might be missing from examples in the training dataset; c) the subset of missing data may not be known in advance, and it could also be randomly distributed across input data. Importantly, we are interested in deriving a method whereby the distribution of the generated data faithfully reflects the Bayesian perspective. Our derivations heavily rely on the Schur complement properties in the spirit of the conditional Gaussian distributions. To highlight this feature we call our approach *VISCOS Flows: VarIational Schur COnditional Sampling with normalizing Flows*. The use of a variational posterior brings some advantages: for example, with a single fitted posterior, multiple samples can quickly be recovered. We also show how to amortise the cost of inference across multiple partially observed items by using inference networks (Kingma & Welling, 2013).

## 2 VISCOS FLOWS: VARIATIONAL SCHUR CONDITIONAL SAMPLING WITH NORMALIZING FLOWS

### 2.1 PRELIMINARIES AND PROBLEM FORMULATION

**Preliminaries**: Consider a $C^1$-diffeomorphism $f(\boldsymbol{X}) : \mathcal{X} \to \mathcal{Y}$ where $\mathcal{X} \subseteq \mathbb{R}^d$, $\mathcal{Y} \subseteq \mathbb{R}^d$ are open sets and define the inverse of $f$ to be $g(\boldsymbol{y}) \equiv f^{-1} : \mathcal{Y} \to \mathcal{X}$. Consider the case where the random variable $\boldsymbol{Y} = f(\boldsymbol{X})$ while $\boldsymbol{X}$ is distributed according to the base distribution $P_0(\boldsymbol{X})$ with density $p_0(\boldsymbol{x})$. According to the change of variable rule $\boldsymbol{Y}$ has a log-density given by the following formula:

$$
\begin{aligned}
\log p(\boldsymbol{y}) &= \log p_0(g(\boldsymbol{y})) \quad + \log |\det \nabla_{\boldsymbol{y}} g(\boldsymbol{y})| \\
&= \log p_0(\boldsymbol{x}) \quad\quad - \log |\det \nabla_{\boldsymbol{x}} f(\boldsymbol{x})|.
\end{aligned}
\tag{1}
$$

We will use two set of complementary indexes covering $[d]$: observable $O \subset [d]$ with cardinality $\#O = d^O$, and hidden $H \subset [d]$ with $\#H = d^H$ and $O \cup H = [d]$. In what follows, without loss of

generality we assume $O = \{0, \ldots, d^O - 1\})$ and $H = \{d^O, \ldots, d\}$, and use the following notation:

$$f(\boldsymbol{x}) = \begin{pmatrix} f^O(\boldsymbol{x}^O; \boldsymbol{x}^H) \\ f^H(\boldsymbol{x}^H; \boldsymbol{x}^O) \end{pmatrix}, \quad \nabla_{\boldsymbol{x}} f(\boldsymbol{x}) = \boldsymbol{J}(\boldsymbol{x}) = \begin{pmatrix} \boldsymbol{J}^{OO}(\boldsymbol{x}) & \boldsymbol{J}^{OH}(\boldsymbol{x}) \\ \boldsymbol{J}^{HO}(\boldsymbol{x}) & \boldsymbol{J}^{HH}(\boldsymbol{x}) \end{pmatrix},$$

and we will use a similar partition for $g(\boldsymbol{y})$ and its Jacobian $\boldsymbol{G}(\boldsymbol{y})$. We will use the notation $m^{OO}(\boldsymbol{J}) = \boldsymbol{J}^{OO}$ to denote sub-matrix masking. The central to our approach is the Schur complements and its properties and, in particular, the following well-known identities:

$$\boldsymbol{G}^{HH}(f(\boldsymbol{x})) = \left( \boldsymbol{J}^{HH}(\boldsymbol{x}) - \boldsymbol{J}^{HO}(\boldsymbol{x}) \left( \boldsymbol{J}^{OO}(\boldsymbol{x}) \right)^{-1} \boldsymbol{J}^{OH}(\boldsymbol{x}) \right)^{-1}, \qquad (2)$$

$$\det(\boldsymbol{J}) = \det\left( \boldsymbol{J}^{OO} \right) \det\left( \boldsymbol{J}^{HH} - \boldsymbol{J}^{HO} \left( \boldsymbol{J}^{OO} \right)^{-1} \boldsymbol{J}^{OH} \right) = \frac{\det\left( \boldsymbol{J}^{OO} \right)}{\det\left( \boldsymbol{G}^{HH} \right)}. \qquad (3)$$

We denote by $\boldsymbol{A}$ the "detached" version of a matrix-valued function $\boldsymbol{A}(\boldsymbol{x})$, i.e. a matrix whose values match those of $\boldsymbol{A}$ but have a null gradient.

**Problem formulation**: Consider a set $\mathcal{Y}$ containing some samples of interest, for example, images. Assume also a pre-trained normalizing flow, i.e., a map $f : \mathcal{X} \to \mathcal{Y}$ with $\mathcal{X}, \mathcal{Y} \subseteq \mathbb{R}^d$. In what follows we relax the assumption on the pre-trained flow, but for streamlining the presentation we assume that the normalizing flow is given at this point. Let us now assume that the observation sample $\boldsymbol{y}$ is partially missing or masked (for instance, photographer's finger is covering part of the image). In particular, the observation is split into observed $\mathcal{Y}^O = \{\boldsymbol{y}^O \mid \boldsymbol{y} \in \mathcal{Y}\}$ and hidden $\mathcal{Y}^H = \{\boldsymbol{y}^H \mid \boldsymbol{y} \in \mathcal{Y}\}$ regions, where $O, H$ are observed and hidden indexes, respectively. **Our goal is to sample $\boldsymbol{y}^H$ from the conditional distribution over a partial observation $\boldsymbol{Y}^O \in \mathcal{Y}^O$.** While this distribution has a density given by $p(\boldsymbol{y}^H \mid \boldsymbol{y}^O) = \frac{p(\boldsymbol{y})}{p(\boldsymbol{y}^O)}$, computing $p(\boldsymbol{y}^O) = \int p(\boldsymbol{y}^O, \boldsymbol{y}^H) d\boldsymbol{y}^H$ is in general intractable. Since it is often easier to sample from and optimise in the latent space we will aim to express all distributions in the space $\mathcal{X}$. First, we introduce a partition into observed and hidden indexes in the latent space $\mathcal{X}$.

**Assumption A0.** Both spaces $\mathcal{X}$ and $\mathcal{Y}$ are partitioned as $O = \{0, \ldots, d^O - 1\}$ and $H = \{d^O, \ldots, d\}$.

We stress that **Assumption A0** is made *only* to simplify the exposition. In general, we can have arbitrary partitions of the observed space, which will not affect our algorithm. While this partition seems somewhat artificial, it enables the use of the Schur complement in our derivations. Choosing the same partition in $\mathcal{X}$ and $\mathcal{Y}$ is natural for the invertible residual networks (iResNet) (Behrmann et al., 2019; Chen et al., 2019) due the

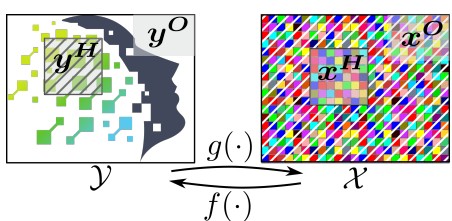

Figure 1: Example of partitioning.

structure of the flow. This observation also holds for any flows sharing a residual structure $\boldsymbol{x} + h(\boldsymbol{x})$ with an activation $h$ (e.g. ReLU), such as Planar and Radial flows (Rezende & Mohamed, 2015), Sylvester flows (van den Berg et al., 2018), continuous flows (Chen et al., 2018; Onken et al., 2021; Mathieu & Nickel, 2020) or – to some extent – Implicit NFs (Lu et al., 2021), where at each layer $t$ of the flow, the derivative of the output $x_{t+1}^i$ with respect to the input layer $\boldsymbol{x}_t$ will be dominated by the $i^{\text{th}}$ component of this input. We note that the partition in latent space $\mathcal{X}$ can be made different from the partition in the observed $\mathcal{Y}$ for numerical purposes. For example, picking a latent space partitioning with a maximal value of $\log |\det(\boldsymbol{J}^{OO})|$ will ensure that the matrix $\boldsymbol{J}^{OO}$ is not ill-conditioned. Indeed, for any matrix $\boldsymbol{A}$ we have $2 \log |\det(\boldsymbol{A})| = \log(\det(\boldsymbol{A}\boldsymbol{A}^T)) = \sum_i \log(\sigma_i(\boldsymbol{A}))$, where $\sigma_i$ are the singular values of $\boldsymbol{A}$. Thus promoting larger singular values will improve conditioning. We empirically found that across several architectures (iResNet, Planar, Radial, Sylvester and Implicit flows), this approach leads to a well-behaving algorithm. Maximising $\log |\det(\boldsymbol{J}^{OO})|$ to choose partitioning can also potentially be used for coupling flow architectures (e.g. Glow (Kingma & Dhariwal, 2018) or RealNVPs (Dinh et al., 2016)), however, this requires additional research. While **Assumption A0** is made for convenience of notation, the following assumption on $f$ is *necessary*.

**Assumption A1.** The map $f : \mathcal{X} \to \mathcal{Y}$ and its restriction $f^O(\boldsymbol{X}^O; \boldsymbol{X}^H) : \mathcal{X}^O \to \mathcal{Y}^O$ for all $\boldsymbol{x}^H \in \mathcal{X}^H$ are $C^1$-diffeomorphisms.

Let us now formally define our reparametrisation from $\mathcal{Y}$ space to $\mathcal{X}$ as follows:

$$\overline{\boldsymbol{x}^H}(\boldsymbol{y}^H, \boldsymbol{y}^O) := g^H(\boldsymbol{y}^H; \boldsymbol{y}^O), \tag{4}$$

$$\overline{\boldsymbol{x}^O}(\boldsymbol{x}^H, \boldsymbol{y}^O) \text{ such that } f^O(\overline{\boldsymbol{x}^O}; \boldsymbol{x}^H) - \boldsymbol{y}^O = 0. \tag{5}$$

Our reparametrisation is well-posed as stated in the following lemma with the proof in Appendix A.

**Lemma 1** *If **Assumption A1** holds then the reparametrisation in Equations 4 and 5 is well-posed, i.e, $g^H(\boldsymbol{y}^H, \boldsymbol{y}^O)$ is a diffeomorphism, while the map $\overline{\boldsymbol{x}^O}(\boldsymbol{x}^H, \boldsymbol{y}^O)$ exists and it is differentiable.*

Now we can rewrite Equation 1 using Equation 3 in simpler terms:

$$\log p(\boldsymbol{y}) = \log p_0(\boldsymbol{x}^H) + \log \left|\det \boldsymbol{G}^{HH}\right| + \log p_0(\boldsymbol{x}^O) - \log \left|\det \boldsymbol{J}^{OO}\right|. \tag{6}$$

Here we used the fact that the base distribution $p_0(\boldsymbol{x})$ can be factorised as a product of independent univariate distributions, and hence $p_0(\boldsymbol{x}) = p_0(\boldsymbol{x}^O)p_0(\boldsymbol{x}^H)$. Now we can define the variational distribution in terms of the latent samples as follows: $q(\boldsymbol{y}^H; \boldsymbol{y}^O) = q(\overline{\boldsymbol{x}^H})\left|\det \boldsymbol{G}^{HH}\right|$. Combining Equation 6 with our variational distributions we obtain the evidence lower bound:

$$\mathcal{L}_{\text{ELBO}} = \mathbb{E}_{q(\boldsymbol{y}^H; \boldsymbol{y}^O)}\left[\log p(\boldsymbol{y}^H, \boldsymbol{y}^O)\right] + \mathbb{H}\left[q(\boldsymbol{y}^H; \boldsymbol{y}^O)\right]$$

$$= \mathbb{E}_{q(\overline{\boldsymbol{x}^H})}\left[\log \frac{p_0(\overline{\boldsymbol{x}^H})}{q(\overline{\boldsymbol{x}^H})} + \log p_0(\overline{\boldsymbol{x}^O}) - \log \left|\det \boldsymbol{J}^{OO}(\overline{\boldsymbol{x}^O}, \overline{\boldsymbol{x}^H})\right|\right], \tag{7}$$

where $\mathbb{H}[\cdot]$ is the entropy and we have taken advantage of the fact that the log-absolute determinant (LAD) of the Jacobian belonging to the approximate posterior $q(\boldsymbol{y}^H; \boldsymbol{y}^O)$ cancels with the one from the joint log-probability given by the model (i.e., the terms with $\boldsymbol{G}^{HH}$ cancel each other out). Furthermore, we have eliminated the dependence on the variable $\boldsymbol{y}^H$ completely instead we treat $\boldsymbol{x}^H$ as a variational variable. We note here that our, at the first sight, artificial partition of the latent space into observable $\boldsymbol{x}^O$ and hidden $\boldsymbol{x}^H$ parts was the key allowing for the reparametrisation from $\mathcal{Y}$ space to $\mathcal{X}$ space. In particular, we used the invertibility of sub-jacobians $\boldsymbol{G}^{HH}$ and $\boldsymbol{J}^{OO}$ allowing to map between observable and hidden samples in the $\mathcal{X}$ and $\mathcal{Y}$ spaces. Our approach is in the spirit of Gaussian conditional sampling and Schur complement, which justifies the method's name. Note that this ELBO loss can be added to other losses, e.g., to a classifier loss as we do in what follows. Note that this ELBO can be used in combination with other losses. Thus the normalizing flow can be trained simultaneously while performing conditional sampling of missing data. This way we can treat masked data during training as well as after training.

There are still two technical issues that we cover in the following subsections before presenting our algorithm. First, we need to get $\overline{\boldsymbol{x}^O}$ by solving Equation 5. Second, we need to discuss the computation of the ELBO gradient with respect to $\overline{\boldsymbol{x}^H}$. Specifically, we need to reparametrise $\overline{\boldsymbol{x}^O}$ and compute the LAD gradient. As the following two sections are quite technical the reader interested in the algorithm itself can skip to Subsection 2.4.

## 2.2 SOLVING THE EQUALITY CONSTRAINT

We consider two root-finding approaches: a new fixed-point iterative algorithm and Newton-Krylov methods. In practice, we used the latter as a fallback option when the former did not converge below the desired convergence threshold.

**Fixed-point iterative algorithm** As our method heavily relies on solving equality constraints, one may wish to reduce the computational burden of this operation. We therefore seek for an efficient gradient-free method that can achieve this in some settings. In particular, we solve simultaneously for $\boldsymbol{x}^O$ and $\boldsymbol{y}^H$ the equations $\boldsymbol{x}^O = g^O(\boldsymbol{y}^O, \boldsymbol{y}^H)$ and $\boldsymbol{y}^H = f^H(\boldsymbol{x}^O, \boldsymbol{x}^H)$, instead of solving $\boldsymbol{y}^O = f^O(\boldsymbol{x}^O, \boldsymbol{x}^H)$ for $\boldsymbol{x}^O$. This allows to consider a fixed-point iterative approach in Algorithm 1, whose convergence properties are analysed in Theorem 1 proved in Appendix B:

**Theorem 1** *Consider Algorithm 1, assume that there exists a unique solution $\widetilde{\boldsymbol{x}^O} = g^O(\boldsymbol{y}^O, \widetilde{\boldsymbol{y}^H})$, $\widetilde{\boldsymbol{y}^H} = f^H(\widetilde{\boldsymbol{x}^O}, \boldsymbol{x}^H)$ around which $g^O$ and $f^H$ are continuously-differentiable, and the global Lipschitz constants of $f^H$ and $g^O$ are equal to $L_a$ and $L_b$, respectively. Then the algorithm converges to the unique solution for any $\alpha \in (0, 1]$, $\beta \in (0, 1]$, if $L_a L_b < 1$.*

---

**Algorithm 1:** Iterative algorithm solving for $\boldsymbol{x}^O$

---

1  **Input:** Invertible flow $f = g^{-1}$, partial (masked) observation $\boldsymbol{y}^O = m^O(\boldsymbol{y})$, conditional latent hidden sub-vector $\boldsymbol{x}^H$, mixing coefficients $\{\alpha, \beta\} \in (0, 1]$

2  **Output:** $\boldsymbol{y}^H$ and $\boldsymbol{x}^O$ satisfying $g^O(\boldsymbol{y}^O, \boldsymbol{y}^H) = \boldsymbol{x}^O$, $f^H(\boldsymbol{x}^O, \boldsymbol{x}^H) = \boldsymbol{y}^H$ for given $\boldsymbol{x}^H$ and $\boldsymbol{y}^O$.

Initialisation: $\boldsymbol{\chi}^O = 0$;

$\widetilde{\boldsymbol{y}^H} = f^H(\boldsymbol{\chi}^O, \boldsymbol{x}^H)$;

$\widetilde{\boldsymbol{x}^O} = \boldsymbol{x}^O = g^O(\boldsymbol{y}^O, \boldsymbol{y}^H)$;

**while** *has not converged* **do**

  $\quad \boldsymbol{y}^H = f^H(\widetilde{\boldsymbol{x}^O}, \boldsymbol{x}^H)$;

  $\quad \widetilde{\boldsymbol{y}^H} = \alpha \boldsymbol{y}^H + (1 - \alpha)\widetilde{\boldsymbol{y}^H}$;

  $\quad \boldsymbol{x}^O = g^O(\boldsymbol{y}^O, \widetilde{\boldsymbol{y}^H})$;

  $\quad \widetilde{\boldsymbol{x}^O} = \beta \boldsymbol{x}^O + (1 - \beta)\widetilde{\boldsymbol{x}^O}$;

**end**

---

**Newton-Krylov methods (Knoll & Keyes, 2004)**: In this approach, the value $\boldsymbol{x}_k^O$ is updated in an iterative manner for a given step size $s$

$$\boldsymbol{x}_{k+1}^O = \boldsymbol{x}_k^O - s\left(\boldsymbol{J}^{OO}\right)^{-1}\left(f^O(\boldsymbol{x}_k^O; \boldsymbol{x}^H) - \boldsymbol{y}^O\right),$$

which is derived using the first order expansion of the flow restriction. The reliance of this method on the inverse Jacobian-vector product $(\boldsymbol{J}^{OO})^{-1}\boldsymbol{u}$ requires us to invert a sub-matrix (defined by the input and output masks) that, in most cases, is only accessible via left or right vector-matrix product, due to the constraints imposed by the use of backward propagation algorithms. Moreover, explicit computation of this sub-Jacobian is also usually prohibitively expensive to retrieve in closed form. Therefore, to compute this product involving an inverse sub-Jacobian, we rely on the generalized minimal residual method (GMRES) (Saad & Schultz, 1986), a Krylov subspace method (Simoncini & Szyld, 2007). At their core, Newton-Krylov methods (Knoll & Keyes, 2004) rely on the computational amortisation of the GMRES-based inversion step by caching intermediate values between successive iterations of the solver (Baker et al., 2005) accelerating the algorithm convergence.

Still, computing a single Jacobian vector product can be quite computationally expensive, for instance when using invertible residual networks (Chen et al., 2019; Behrmann et al., 2019). To this end we employ an identity reciprocal to Equation 2, i.e., $(\boldsymbol{J}^{OO})^{-1} = \boldsymbol{G}^{OO} - \boldsymbol{G}^{OH}(\boldsymbol{G}^{HH})^{-1}\boldsymbol{G}^{HO}$, giving an alternative representation of $(\boldsymbol{J}^{OO})^{-1}$. First, if $d^O \gg d^H$, then inverting the sub-Jacobian $\boldsymbol{G}^{HH}$ is more computationally efficient, e.g., using GMRES or a similar technique. In this case, the inverse of the sub-Jacobian can be obtained or approximated only by querying Jacobian-vector products involving $\boldsymbol{G}$, and not $\boldsymbol{J}$. Otherwise, when computing the plain inverse of $\boldsymbol{J}^{OO}$ using GMRES, we used $\boldsymbol{G}^{OO}$ as a preconditioner. We found this to speed up the computation by a factor of 2 to 4, depending on the size of the problem and the architecture used.

## 2.3  ELBO GRADIENT

Taking the gradient of ELBO relies on solving two technical issues: computing the gradient of $\overline{\boldsymbol{x}^O}$ and the gradient of LAD with respect to variational posterior parameters $\boldsymbol{\theta}$.

**Reparametrising** $\overline{\boldsymbol{x}^O}$**.** According to the implicit function theorem and Equation 5, we can reparameterise $\boldsymbol{x}^O$ as a function of $\boldsymbol{x}^H$, $\boldsymbol{y}^O$ and $\boldsymbol{\theta}$ to:

$$\nabla_{\boldsymbol{x}^H}\overline{\boldsymbol{x}^O} = -(\boldsymbol{J}^{OO})^{-1}\boldsymbol{J}^{OH}.$$

The inverse of the sub-Jacobian can be obtained via GMRES algorithm with similar considerations regarding methods to speed up this computation to the Newton-Krylov method in Section 2.2. Reparameterising as a function of the variational posterior parameters $\boldsymbol{\theta}$ is then trivially achieved via pathwise or implicit gradient calculation.

**Log-Absolute Determinant gradient Estimators.** First recall that the gradient of the log-determinant can be represented using trace as follows (Petersen & Pedersen, 2008):

$$\nabla_{\boldsymbol{x}}\log|\det \boldsymbol{A}(\boldsymbol{x})| = \nabla_{\boldsymbol{x}}\text{Tr}\left[\boldsymbol{A}^{-1}\boldsymbol{A}(\boldsymbol{x})\right],$$

---

**Algorithm 2:** VISCOS Flows: Variational Schur Conditional Sampling with Normalizing Flows

---

1 **Input:** Base distribution $p_0(\boldsymbol{x})$, invertible flow $f = g^{-1} : \mathcal{X} \to \mathcal{Y}$, partial (masked)
    observation $\boldsymbol{y}^O = m^O(\boldsymbol{y})$
2 **Output:** Conditional sample $\boldsymbol{y}^H \sim p(\cdot \mid \boldsymbol{y}^O)$
  Initialization: set $q_{\boldsymbol{\theta}}(\boldsymbol{x}^H)$, select partitioning function $\boldsymbol{x}^O - m^O(\boldsymbol{x})$;
  **while** *has not converged* **do**
  $\quad$ sample $\boldsymbol{x}^H \sim q_{\boldsymbol{\theta}}(\boldsymbol{x}^H)$;
  $\quad$ compute $\overline{\boldsymbol{x}^O}$ as in Section 2.2;
  $\quad$ reparameterise $\overline{\boldsymbol{x}^O}(\boldsymbol{x}^H, \boldsymbol{y}^O, \boldsymbol{\theta})$;
  $\quad$ compute stochastic gradient estimate $\nabla_{\boldsymbol{\theta}}\mathcal{L}(q)$ and update variational posterior;
  **end**

---

which allows us to use built-in functions if the inverse of $\boldsymbol{A}$ is readily available. Hence, although the value of $\log|\det \boldsymbol{A}(\boldsymbol{x})|$ cannot be estimated in closed form, its gradient only requires us to differentiate the trace of the product of $\boldsymbol{A}^{-1}$ and $\boldsymbol{A}(\boldsymbol{x})$, where only the latter is differentiated through.

In many cases, one learns the map $g(\cdot)$ and hence $\boldsymbol{J}$ is only implicitly defined by its inverse $\boldsymbol{G}$ (e.g. $\boldsymbol{J}$ is obtained through Neumann series or Krylov methods). In such cases, higher order derivatives (i.e., $\nabla_{\boldsymbol{x}}\boldsymbol{J}(\boldsymbol{x})$) cannot be accessed readily by backpropagating through the graph. Therefore, it is necessary to express $\boldsymbol{J}(\boldsymbol{x})$ through $\boldsymbol{G}(\boldsymbol{y})$. In particular, if $d^O \ll d^H$ we can obtain the Natural LAD gradient Estimator (NLADE) of the sub-Jacobian $\log|\det \boldsymbol{J}^{OO}(\boldsymbol{x}^H)|$:

$$\nabla_{\boldsymbol{x}} \log|\det \boldsymbol{J}^{OO}(\boldsymbol{x})| = \nabla_{\boldsymbol{x}}\mathrm{Tr}\left[(\boldsymbol{J}^{OO})^{-1}\boldsymbol{J}^{OO}(\boldsymbol{x})\right] =$$
$$\nabla_{\boldsymbol{y}}\mathrm{Tr}\left[(\boldsymbol{J}^{OO})^{-1}m^{OO}\left(\boldsymbol{G}^{-1}\boldsymbol{G}(\boldsymbol{y})\boldsymbol{G}^{-1}\right)\right]\Big|_{\boldsymbol{y}=f(\boldsymbol{x})}\boldsymbol{J}(\boldsymbol{x}), \quad (8)$$

where we used $\nabla_{\boldsymbol{x}}(\boldsymbol{J}(\boldsymbol{x})) = \nabla_{\boldsymbol{x}}(\boldsymbol{G}(f(\boldsymbol{x})))^{-1} = \nabla_{\boldsymbol{x}}(\boldsymbol{G}^{-1}\boldsymbol{G}(f(\boldsymbol{x}))\boldsymbol{G}^{-1}) = \nabla_{\boldsymbol{y}}(\boldsymbol{G}^{-1}\boldsymbol{G}(\boldsymbol{y})\boldsymbol{G}^{-1})\nabla_{\boldsymbol{x}}f(\boldsymbol{x})$ (Petersen & Pedersen, 2008). Using NLADE can be inefficient if $d^H \ll d^O$. In this case, one could express $(\boldsymbol{J}^{OO})^{-1} = \boldsymbol{G}^{OO} - \boldsymbol{G}^{OH}(\boldsymbol{G}^{HH})^{-1}\boldsymbol{G}^{HO}$, but this approach can be executed more efficiently using the identity $\log|\det \boldsymbol{J}^{OO}(\boldsymbol{x})| = \log|\det \boldsymbol{G}^{HH}(f(\boldsymbol{x}))| - \log|\det \boldsymbol{G}(f(\boldsymbol{x}))|$ in the first place. This leads to the Corrected LAD gradient Estimator (CLADE):

$$\nabla_{\boldsymbol{x}} \log|\det \boldsymbol{J}^{OO}(\boldsymbol{x})| = \nabla_{\boldsymbol{y}}\left(\mathrm{Tr}\left[(\boldsymbol{G}^{HH})^{-1}\boldsymbol{G}^{HH}(\boldsymbol{y})\right] - \mathrm{Tr}\left[\boldsymbol{G}^{-1}\boldsymbol{G}(\boldsymbol{y})\right]\right)\Big|_{\boldsymbol{y}=f(\boldsymbol{x})}\boldsymbol{J}(\boldsymbol{x}). \quad (9)$$

Both expressions have their advantages and possible drawbacks: if $d^O \ll d^H$, then NLADE would be more efficient. On the other hand, if $d^H \ll d^O$, then CLADE appears to be a better option. Furthermore, as CLADE does not use implicit matrix value $(\boldsymbol{J}^{OO})^{-1}$ (defined through $\boldsymbol{G}$), its variance is potentially lower than NLADE's. In other cases, the choice would depend on the efficiency of matrix inversions of $\boldsymbol{J}, \boldsymbol{G}, \boldsymbol{J}^{OO}, \boldsymbol{G}^{HH}$.

## 2.4 ALGORITHM

We suggest a two-pass iterative algorithm aimed at sampling from a conditional normalizing flow using a parametric posterior $q_{\boldsymbol{\theta}}(\boldsymbol{y}^H \mid \boldsymbol{y}^O)$: first, a set of conditional samples is drawn by solving the equations $g^O(\boldsymbol{y}^O, \boldsymbol{y}^H) - \boldsymbol{x}^O = 0$, $f^H(\boldsymbol{x}^O, \boldsymbol{x}^H) - \boldsymbol{y}^H = 0$ or $f^O(\boldsymbol{x}^O, \boldsymbol{x}^H) - \boldsymbol{y}^O = 0$. Next, this sample is reparameterised given the variational posterior parameters, and lastly an optimisation step is made given the stochastic estimate of the ELBO gradient. The summary is given in Algorithm 2.

We used two distinct designs of approximate posterior depending on the task at hand. When possible, each data to be completed was treated separately, with a single variational posterior configuration being fitted to each item in the dataset. When the amount of incomplete items was too large to be completed on a one-to-one basis, we amortised the cost of inference by acquiring the variational posterior parameters through a inference network that was trained to minimize the KL divergence in Equation 7. More details can be found in Appendix C.

## 3 EXPERIMENTS

**Experimental setup** We tested our variational conditional sampling algorithm on a classification and data completion task when the training and test data were partially observed, and on a post-training data completion task. Due to the restriction imposed by **Assumption A1**, we only considered the iResNet (Behrmann et al., 2019; Chen et al., 2019) architecture to test our approach on in the main text. Appendix D provides results with Implicit NFs, showing that our approach can be extended to other types of architectures with possibly higher Lipschitz constants. Several steps of our gradient computation rely on computing the inverse of the Jacobian of the transform. This quantity was queried using Neumann series: at each layer $i \in [L]$ of a $L$-deep network, we locally computed the product of a vector with the inverse Jacobian as

$$\boldsymbol{u}\boldsymbol{J}_i^{-1} = \boldsymbol{u}\sum_{j=0}^{\infty} - \left(\nabla_x h(\boldsymbol{x})\right)^j .$$

In practice, the sum was truncated when the difference between two successive iterations was below a predefined threshold $\epsilon$ ($10^{-5}$ for single and $10^{-10}$ for double precision). Jacobian-vector products were computed using the identity $\boldsymbol{J}(\boldsymbol{x})\boldsymbol{u} = \boldsymbol{u}\nabla_{\boldsymbol{v}}\left[\boldsymbol{v}\boldsymbol{J}(\boldsymbol{x})\right]$ for some $\boldsymbol{v}$. As this formula requires a graph to be built on top of another one, and to avoid memory overflow, we computed this quantity locally at each layer of the network. The convergence threshold for Algorithms 1 and 2 was $10^{-3}$.

At early stages of training, Algorithm 1 quickly converged with $\alpha = \beta = 1$. However, lower values were needed later on, or when using a fully trained model (in the post-training data imputation setting). Therefore, we adopted the following schedule throughout the training process: an initial mixing rate was set to $\alpha = \beta = 0.5$, and at each iteration of Algorithm 1, the mixing rate was decayed by a factor of $0.95$. In less than 1% of the iterations, our fixed-point algorithm did not reach the convergence threshold, and then we used a Newton-Krylov solver. In the case of MNIST dataset, each optimisation step took approximately 20 seconds, roughly equally split between solving the equality constraint retrieving the gradient estimate.

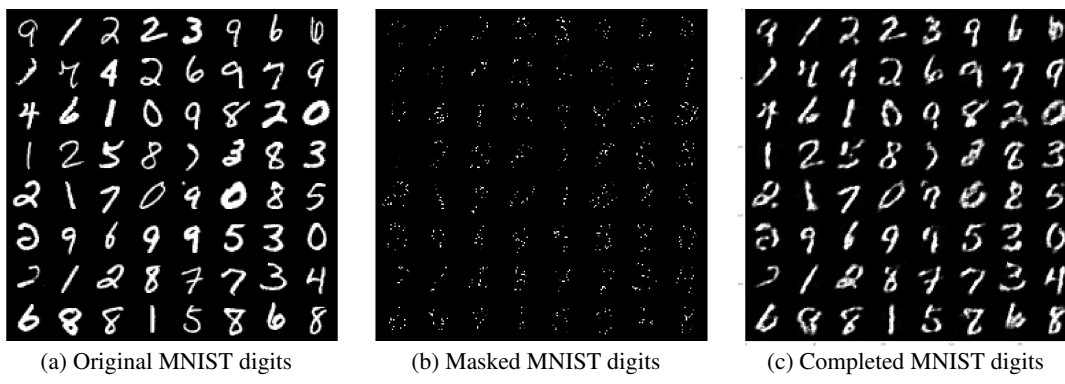

(a) Original MNIST digits      (b) Masked MNIST digits      (c) Completed MNIST digits

Figure 2: MNIST completion with 90% missingness rate.

**Quantitative experiment: Training on incomplete MNIST data** To test the capability of Variational Schur Conditional Sampling (VISCOS Flows) to learn from incomplete data, we used the MNIST dataset (LeCun & Cortes, 2010) where a percentage of the pixels of each digit image was missing. We considered only the challenging setting where the missing pixels were randomly spread across the image. The pre-trained NF component was similar to (Chen et al., 2019) and consisted of an iResNet model with 73 convolutional layers with 128 channels each, followed by four fully connected residual layers. The trained inference network consisted of a 5-layer convolutional network with a $3 \times 3$ kernel, SAME-padding strategy and with $[4, 8, 16, 32, 8]$ channels at each layer, respectively. For this network, the ReLU activation function (Nair & Hinton, 2010) was used. Finally, we used a simple classifier consisting of the sequence of a squeezing layer, a 1-dimensional batch-norm layer (Ioffe & Szegedy, 2015) and a bias-free linear layer. We used the Adam optimiser (Kingma & Ba, 2015) with a learning rate of $10^{-2}$, decaying by a factor of $0.5$ at every epoch. The batch size was set to 64 digits, and the components were trained simultaneously for a total of 5 epochs. The NLADE gradient estimator was used for these experiments with the Hutchinson stochastic trace

| Missing rate → | 0.5 | 0.6 | 0.7 | 0.8 | 0.9 |
|---|---|---|---|---|---|
| MisGAN | 0.968 | 0.945 | 0.872 | 0.690 | 0.334 |
| PL-MCMC | - | - | - | - | - |
| MCFlow | **0.985** | **0.979** | **0.963** | 0.905 | 0.705 |
| VISCOS Flow (ours) | 0.9616 | 0.9588 | 0.9485 | **0.9355** | **0.886** |

Table 1: Top-1 classification accuracy on incomplete MNIST dataset (higher is better). We highlight in bold the best performance, while the symbol "-" represents a missing experiment.

| Missing rate → | 0.5 | 0.6 | 0.7 | 0.8 | 0.9 |
|---|---|---|---|---|---|
| MisGAN | 0.3634 | 0.8870 | 1.324 | 2.334 | **6.325** |
| PL-MCMC | - | 5.7 | - | - | 87 |
| MCFlow | 0.8366 | 0.9082 | 1.951 | 6.765 | 15.11 |
| VISCOS Flow (ours) | **0.2692** | **0.4398** | **0.7823** | **1.5491** | 7.315 |

Table 2: FID on incomplete MNIST dataset (lower is better). We highlight in bold the best performance, while the symbol "-" represents a missing experiment.

| Missing rate → | 0.5 | 0.6 | 0.7 | 0.8 | 0.9 |
|---|---|---|---|---|---|
| MisGAN | 0.12174 | 0.13393 | 0.15445 | 0.19455 | 0.27806 |
| PL-MCMC | - | 0.1585 | - | - | 0.261 |
| MCFlow | **0.10045** | **0.11255** | **0.12996** | 0.15806 | 0.20801 |
| VISCOS Flow (ours) | 0.1127 | 0.1221 | 0.1340 | **0.1470** | **0.1924** |

Table 3: RMSE on incomplete MNIST dataset (lower is better). We highlight in bold the best performance, while the symbol "-" represents a missing experiment.

estimator (Hutchinson, 1990), taking a single sample for each gradient computation. We compared our results to the ones presented in PL-MCMC (Cannella et al., 2020), MCFlow (Richardson et al., 2020) and MisGAN (Li et al., 2019). Due to the limits of our computational resources, we resort to the performances displayed in (Richardson et al., 2020; Cannella et al., 2020). All the presented results are computed on MNIST official test dataset, which was not used for training. Figure 2 gives a visual account of the performance of the VISCOS Flows data completion capability. In all cases, VISCOS Flows was either first or second ranked in terms of Top-1 classification accuracy (Table 1), Fréchet Inception Score (FID (Heusel et al., 2017), Table 2) or Root-Mean-Squared-Error (RMSE, Table 3). It is important to note that, although MCFlow outperformed our approach on some occasions, this algorithm relies on the assumption that one can train a separate classifier on complete data, and then use this classifier on the completed dataset. Our classifier was, instead, trained on-the-fly on the incomplete dataset only.

**Qualitative experiment: post-training data imputation** We now turn to the problem of completing a partially observed sample using the VISCOS Flow technique. We trained a standard iResNet-164 architecture on the CIFAR-10 dataset (Krizhevsky, 2009) for 250 epochs. VISCOS Flow completion capability was compared to PL-MCMC, as MCFlow cannot perform post-training data imputation. The task was similar to the one presented in (Cannella et al., 2020): a masked centered square of size $8 \times 8$ had to be completed. PL-MCMC completion was achieved using the same hyperparameters provided by the authors. For VISCOS Flows, we used a Gaussian approximate posterior with a sparse approximation to the covariance matrix encoded by 50 Householder layers. The parameters were trained using the Adam optimiser with a learning rate of $10^{-2}$, and at each step, a batch of 8 completed images were generated to obtain a gradient estimate. A total of 500 steps showed to be sufficient to reach convergence. Figure 3 shows how our algorithm compared with PL-MCMC for a set of test images. We also included the first solution obtained by the initial standard Gaussian distribution to show that, although training did make the images look more convincing, the initial guess was already somewhat well fitted to the rest of the image. As a qualitative measure of fitness, we measured the RMSE between generated and true images for both algorithms, and obtained a similar value for both (VISCOS Flow (NLADE): $0.2291 \pm 0.1570$, PL-MCMC: $0.2287 \pm 0.1961$, where the confidence interval is $0.95\%$ of standard deviation). By comparison, at the first iteration of the VISCOS Flow algorithm, an RMSE of $0.2802 \pm 0.1188$ was measured. As the gradient of the variational posterior parameters can be retrieved with two different methods (CLADE or NLADE), we compared one against the other, but observed little qualitative or quantitative difference between the two (RMSE for VISCOS Flow (CLADE): $0.2320 \pm 0.1793$). A noticeable advantage of VISCOS Flow is that with a single fit of the variational posterior can generate a wide variety of candidate

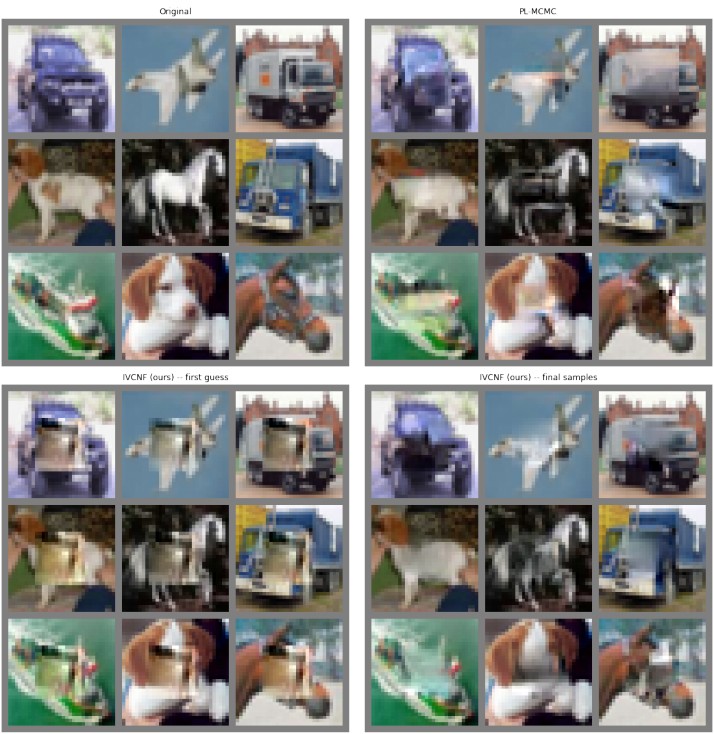

Figure 3: Upper row: Original images (left panel) and PL-MCMC reconstruction (right panel). Lower row: VISCOS Flow after the first iteration (left panel), VISCOS Flow after the final iteration

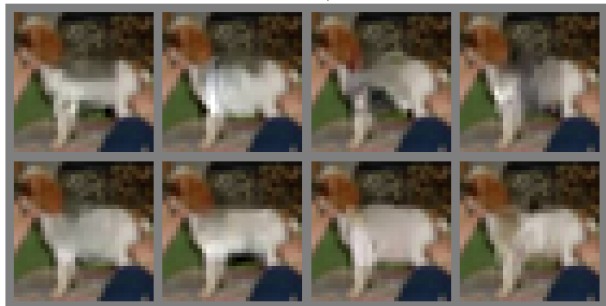

Figure 4: Random samples obtained from a VISCOS Flow trained variational posterior.

images with little effort, as only a few seconds are needed to solve the equality constraint depicted above. Figure 4 gives some flavour of this capability.

## 4 CONCLUSION

This article presents VISCOS Flows, a new method for data completion with NFs. This technique can be successfully applied to architectures when the units of the flow are made of instances of residual layers. Unlike other approaches, such as MCFlow (Richardson et al., 2020) and ACFlow (Li et al., 2020), we do not need to include the conditioning in the training process. This feature improves algorithm's modularity by allowing to decouple NF training and conditional sampling. While PL-MCMC (Cannella et al., 2020) also enables such modularity, it relies on MCMC algorithms, which are arguably harder to use than our gradient descent based algorithm. For some important classes of NFs (e.g. composed of coupling flows, such as Glow (Kingma & Dhariwal, 2018) or RealNVP (Dinh et al., 2016)) the partitioning choice and solving for $x^O$ may be more involved than for NFs with residual architectures. Future work should focus on finding alternative ways to fit low-dimensional variational posteriors for data completion in those cases.

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

## A   DIFFERENTIABILITY OF THE REPARAMETRISATION

For convenience we state the lemma

**Lemma 1** *If **Assumption A1** holds then the reparametrisation in Equations 4 and 5 is well-posed, i.e, $g^H$ is a diffeomorphism, while the map $\overline{x^O}(x^H, y^O)$ exists and it is differentiable.*

First, let us show that $g^H$ is a $C^1$ diffeormorphism. Recall that we use the notation $J^{OO} = \nabla_{x^O} f^O$, $J^{OH} = \nabla_{x^H} f^O$, $J^{HO} = \nabla_{x^H} f^O$, $J^{HH} = \nabla_{x^H} f^H$. Note that the matrices $J^{OO}$, $J$ are invertible since $f(\cdot)$ and $f^O(\cdot; x^H)$ for all $x^H$ are $C^1$-diffeomorphisms. Now, according to the Guttman rank additivity formula (Guttman, 1946) if $J^{OO}$ is an invertible submatrix of an invertible matrix $J$, then

$$\text{rank}(J) = \text{rank}(J^{OO}) + \text{rank}\left(J^{HH} - J^{HO}\left(J^{OO}\right)^{-1} J^{OH}\right).$$

It is easy to see that in this case, the block $J^{HH} - J^{HO}\left(J^{OO}\right)^{-1} J^{OH}$ is also full-rank, and its inverse is given by $G^{HH}$ which is the Jacobian matrix $\nabla_{y^H} g^H(y^H; y^O)$. As these identities are valid for all $x$, the Jacobian matrix of $g^H(y^H; y^O)$ for $y^O$ is always invertible and hence $g^H$ is a $C^1$-diffeomorphism, as well.

Now let us show the second part of the statement. Since $f^O(\overline{x^O}; x^H)$ is a diffeomorphism, due to implicit function theorem there exists a locally differentiable map $\overline{x^O}(x^H, y^O)$ for all $x^H, y^O$.

## B   THEOREM 1: A CONVERGENCE RESULT

For convenience we restate the theorem.

**Theorem 1** *Consider Algorithm 1, assume that there exists a unique solution $\widetilde{x^O} = g^O(y^O, \widetilde{y^H})$, $\widetilde{y^H} = f^H(\widetilde{x^O}, x^H)$ around which $g^O$ and $f^H$ are continuously-differentiable, and the global Lipschitz constants of $f^H$ and $g^O$ are equal to $L_a$ and $L_b$, respectively. Then the algorithm converges to the unique solution for any $\alpha \in (0, 1]$, $\beta \in (0, 1]$, if $L_a L_b < 1$.*

We argue convergence of the iterative procedure using dynamical systems theory and the notion of stability (cf., (Sontag, 2013)). In general, stability is a stronger property than convergence, as stability requires convergence under perturbations in the initial conditions of an iterative procedure. We will prove the theorem in Proposition 1, in what follows, but first we need to develop theoretical notions of stability.

Recall that the updates in the algorithm are as follows:

$$\widetilde{y^H} := \alpha f^H(\widetilde{x^O}, x^H) + (1 - \alpha)\widetilde{y^H}$$
$$\widetilde{x^O} := \beta g^O(y^O, \widetilde{y^H}) + (1 - \beta)\widetilde{x^O}$$

By letting $\eta$, $\xi$ stand for $\widetilde{y^H}$, $\widetilde{x^O}$, respectively, and $a(\cdot) = f^H(\cdot, x^H)$ and $b(\cdot) = g^O(y^O, \cdot)$, our iterative procedure can be written without loss of generality as the following dynamical system:

$$\begin{aligned}
\eta^{k+1} &= (1 - \alpha)\eta^k + \alpha a(\xi^k), \\
\xi^{k+1} &= (1 - \beta)\xi^k + \beta b(\eta^{k+1}).
\end{aligned} \tag{10}$$

We now will review stability results, which are presented for the systems $\zeta^{k+1} = C(\zeta^k)$ with the initial state $\zeta^0 = \nu$, we will denote systems' trajectories as $\zeta^k(\nu)$. In our bounds, we will make use of the class of $\mathcal{KL}$ functions. The function $\gamma \in \mathcal{KL}$ if $\gamma : \mathbb{R}_{\geq 0} \times \mathbb{R}_{\geq 0} \to \mathbb{R}_{\geq 0}$, $\gamma$ is continuous and strictly increasing in the first argument, continuous and strictly decreasing in the second argument, with $\lim_{t \to \infty} \gamma(\zeta, k) = 0$ for every $\zeta$, $\gamma(0, k) = 0$ for all $k \geq 0$.

**Definition 1** *A system is called globally asymptotically stable at $\zeta^*$ if there exists a function $\gamma \in \mathcal{KL}$ such that for all $\nu$ and all $k$:*

$$\|\zeta^k(\nu) - \zeta^*\| \leq \gamma(\|\nu - \zeta^*\|, k),$$

Stability theory offers many additional tools, for instance, the ability of studying how trajectories behave in comparison to each other, which can be done using contraction theory (Lohmiller & Slotine, 1998) or using a similar concept of incremental stability(Tran et al., 2016).

**Definition 2** *A system is called globally asymptotically incrementally stable if there exists a function $\gamma$ such that for all $\boldsymbol{\nu}_1$, $\boldsymbol{\nu}_2$ and all $k$:*

$$\|\boldsymbol{\zeta}^k(\boldsymbol{\nu}_2) - \boldsymbol{\zeta}^k(\boldsymbol{\nu}_1)\| \leq \gamma(\|\boldsymbol{\nu}_2 - \boldsymbol{\nu}_1\|, k),$$

In our derivation, we will also employ local convergence concepts and results

**Definition 3** *A system is called locally asymptotically stable at $\boldsymbol{\zeta}^*$ if there exist a function $\gamma$ and $\varepsilon > 0$ such that for all $\|\boldsymbol{\nu} - \boldsymbol{\zeta}^*\| < \varepsilon$ and all $k$:*

$$\|\boldsymbol{\zeta}^k(\boldsymbol{\nu}) - \boldsymbol{\zeta}^*\| \leq \gamma(\|\boldsymbol{\nu} - \boldsymbol{\zeta}^*\|, k),$$

Checking local stability is rather straightforward given some regularity conditions. In particular, we need to check the magnitude of the spectral radius of $\partial \boldsymbol{C}(\boldsymbol{\zeta})$. Recall that spectral radius $\rho(\boldsymbol{D})$ is defined as $\rho = \max_i |\lambda_i(\boldsymbol{D})|$, where $\lambda_i$ are the eigenvalues of $\boldsymbol{D}$.

**Lemma 2** *Let $\boldsymbol{A}$ be continuously-differentiable around $\boldsymbol{\zeta}^*$. If the spectral radius $\rho(\partial \boldsymbol{C}(\boldsymbol{\zeta}^*))$ is strictly smaller than one, then the system is locally asymptotically stable around $\boldsymbol{\zeta}^*$. If $\boldsymbol{C}(\boldsymbol{\zeta})$ is a linear function, then the system is also globally stable.*

We also have this lemma linking incremental stability and stability properties, which can be shown in a straightforward manner.

**Lemma 3** *If the system $\boldsymbol{\zeta}^{k+1} = \boldsymbol{C}(\boldsymbol{\zeta}^k)$ is locally asymptotically stable around $\boldsymbol{\zeta}^*$ and globally asymptotically incrementally stable, then it is also globally asymptotically stable.*

Finally, we need the following lemma, which is a combination of two results discussed in (Karow et al., 2006): properties of nonnegative matrices (the property $\rho 2$) and Lemma 4.1., which follows the introduction of these properties.

**Lemma 4** *Let $\boldsymbol{C}_{jk} \in \mathbb{R}^{l_j \times l_k}$, $\|\cdot\|$ is the singular value matrix norm, and $\|\boldsymbol{C}_{jk}\| \leq c_{jk}$ then*

$$\rho\left(\begin{pmatrix} \boldsymbol{C}_{11} & \dots & \boldsymbol{C}_{1m} \\ \vdots & & \vdots \\ \boldsymbol{C}_{m1} & \dots & \boldsymbol{C}_{mm} \end{pmatrix}\right) \leq \rho\left(\begin{pmatrix} \|\boldsymbol{C}_{11}\| & \dots & \|\boldsymbol{C}_{1m}\| \\ \vdots & & \vdots \\ \|\boldsymbol{C}_{m1}\| & \dots & \|\boldsymbol{C}_{mm}\| \end{pmatrix}\right) \leq \rho\left(\begin{pmatrix} c_{11} & \dots & c_{1m} \\ \vdots & & \vdots \\ c_{m1} & \dots & c_{mm} \end{pmatrix}\right).$$

Now we are ready to proceed with the proof.

**Proposition 1** *The system equation 10 is globally asymptotically incrementally stable for any $\alpha, \beta \in (0, 1)$ if $L_a L_b < 1$. Furthermore, if there exists $\boldsymbol{\xi}^*$, $\boldsymbol{\eta}^*$ such that $\boldsymbol{\xi}^* = a(\boldsymbol{\eta}^*)$, $\boldsymbol{\eta}^* = b(\boldsymbol{\xi}^*)$, then the system is globally asymptotically stable.*

**Proof:** First let us show incremental stability. Let $\delta\boldsymbol{\eta}^k = \boldsymbol{\eta}_1^k - \boldsymbol{\eta}_2^k$, $\delta\boldsymbol{\eta}^{k+1} = \boldsymbol{\eta}_1^{k+1} - \boldsymbol{\eta}_2^{k+1}$, $\delta\boldsymbol{\xi}^k = \boldsymbol{\xi}_1^k - \boldsymbol{\xi}_2^k$, and $\delta\boldsymbol{\xi}^{k+1} = \boldsymbol{\xi}_1^{k+1} - \boldsymbol{\xi}_2^{k+1}$, then we have

$$\|\delta\boldsymbol{\eta}^{k+1}\| = \|\boldsymbol{\eta}_1^{k+1} - \boldsymbol{\eta}_2^{k+1}\| = \|(1-\alpha)(\boldsymbol{\eta}_1^k - \boldsymbol{\eta}_2^k) + \alpha(a(\boldsymbol{\xi}_1^k) - a(\boldsymbol{\xi}_2^k))\| \leq$$
$$(1-\alpha)\|\boldsymbol{\eta}_1^k - \boldsymbol{\eta}_2^k\| + \alpha\|a(\boldsymbol{\xi}_1^k) - a(\boldsymbol{\xi}_2^k))\| \leq (1-\alpha)\|\delta\boldsymbol{\eta}^k\| + \alpha L_a\|\delta\boldsymbol{\xi}^k\|.$$

Similarly

$$\|\delta\boldsymbol{\xi}^{k+1}\| = \|\boldsymbol{\xi}_1^{k+1} - \boldsymbol{\xi}_2^{k+1}\| = \|(1-\beta)(\boldsymbol{\xi}_1^k - \boldsymbol{\xi}_2^k) + \beta(b(\boldsymbol{\eta}_1^{k+1}) - b(\boldsymbol{\eta}_2^{k+1}))\| \leq$$
$$(1-\beta)\|\boldsymbol{\xi}_1^k - \boldsymbol{\xi}_2^k\| + \beta\|b(\boldsymbol{\eta}_1^{k+1}) - b(\boldsymbol{\eta}_2^{k+1})\| \leq (1-\beta)\|\delta\boldsymbol{\xi}^k\| + \beta L_b\|\delta\boldsymbol{\eta}^{k+1}\| \leq$$
$$((1-\beta) + \alpha\beta L_a L_b)\|\delta\boldsymbol{\xi}^k\| + (1-\alpha)\beta L_b\|\delta\boldsymbol{\eta}^k\|.$$

Summarising we have the following bounds

$$
\begin{aligned}
\|\delta\boldsymbol{\eta}^{k+1}\| &\le (1-\alpha)\|\delta\boldsymbol{\eta}^k\| + \alpha L_a\|\delta\boldsymbol{\xi}^k\|, \\
\|\delta\boldsymbol{\xi}^{k+1}\| &\le (1-\alpha)\beta L_b\|\delta\boldsymbol{\eta}^k\| + ((1-\beta)+\alpha\beta L_b L_a)\|\delta\boldsymbol{\xi}^k\|,
\end{aligned}
\tag{11}
$$

which we can equivalently write in the matrix form:

$$
\begin{pmatrix} \|\delta\boldsymbol{\eta}^{k+1}\| \\ \|\delta\boldsymbol{\xi}^{k+1}\| \end{pmatrix} \le \begin{pmatrix} 1-\alpha & \alpha L_a \\ (1-\alpha)\beta L_b & (1-\beta)+\alpha\beta L_b L_a \end{pmatrix} \begin{pmatrix} \|\delta\boldsymbol{\eta}^k\| \\ \|\delta\boldsymbol{\xi}^k\| \end{pmatrix}.
\tag{12}
$$

As these bounds are valid for every step $k$, we can study convergence of $\|\delta\boldsymbol{\eta}^k\|$, $\|\delta\boldsymbol{\xi}^k\|$ using the system:

$$
\begin{pmatrix} z_1^{k+1} \\ z_2^{k+1} \end{pmatrix} = \underbrace{\begin{pmatrix} 1-\alpha & \alpha L_a \\ (1-\alpha)\beta L_b & 1-\beta+\alpha\beta L_b L_a \end{pmatrix}}_{C} \begin{pmatrix} z_1^k \\ z_2^k \end{pmatrix}.
$$

Indeed, if $|z_1^k|^2 + |z_2^k|^2$ converges to zero as $k \to \infty$, then $\|\delta\boldsymbol{\eta}^k\|^2 + \|\delta\boldsymbol{\xi}^k\|^2$ also converges to zero as $k \to \infty$. According to Lemma 2 as long as the matrix $C$ has the largest absolute value of the eigenvalues smaller than one, the sequence $\{|z_1^k|^2 + |z_2^k|^2\}$ converges to zero as $k \to \infty$ for any initialization $z_1^0 = \|\delta\boldsymbol{\eta}^0\|$, $z_2^0 = \|\delta\boldsymbol{\xi}^k\|$. This would imply that $\|\delta\boldsymbol{\eta}^k\|^2 + \|\delta\boldsymbol{\xi}^k\|^2$ converges to zero as $k \to \infty$ and hence the system equation 10 is global asymptotic incremental stability.

Now all we need to show is that the spectral radius is strictly smaller than one. First, we make a simple transformation obtained by a change of variables $z_2 = \sqrt{\frac{(1-\alpha)\beta L_b}{\alpha L_a}}\tilde{z}_2$ resulting in the following dynamical system

$$
\begin{pmatrix} z_1^{k+1} \\ \tilde{z}_2^{k+1} \end{pmatrix} = \underbrace{\begin{pmatrix} 1-\alpha & \sqrt{\alpha(1-\alpha)\beta L_a L_b} \\ \sqrt{\alpha(1-\alpha)\beta L_a L_b} & 1-\beta+\alpha\beta L_b L_a \end{pmatrix}}_{\widetilde{C}} \begin{pmatrix} z_1^k \\ \tilde{z}_2^k \end{pmatrix}.
$$

Since the system matrix $\widetilde{C}$ is symmetric, all we have to do is derive the conditions when

$$
\begin{pmatrix} 1-\alpha & \sqrt{\alpha(1-\alpha)\beta L_a L_b} \\ \sqrt{\alpha(1-\alpha)\beta L_a L_b} & 1-\beta+\alpha\beta L_b L_a \end{pmatrix} \prec \begin{pmatrix} 1 & 0 \\ 0 & 1 \end{pmatrix},
$$

which is equivalent to

$$
\begin{pmatrix} \alpha & \sqrt{\alpha(1-\alpha)\beta L_a L_b} \\ \sqrt{\alpha(1-\alpha)\beta L_a L_b} & \beta-\alpha\beta L_a L_b \end{pmatrix} \succ 0 \iff
$$
$$
\alpha\beta - \alpha^2\beta L_a L_b > \alpha(1-\alpha)\beta L_a L_b \iff \alpha\beta > \alpha\beta L_a L_b \iff 1 > L_a L_b.
$$

This proves that $\{|z_1^k|^2 + |\tilde{z}_2^k|^2\}$ and consequently $\|\delta\boldsymbol{\eta}^k\|^2 + \|\delta\boldsymbol{\xi}^k\|^2$ converge to zero as $k \to \infty$.

Now we need to show that the system is locally asymptotically stable around any solution to $\boldsymbol{\xi}^* = a(\boldsymbol{\eta}^*)$, $\boldsymbol{\eta}^* = b(\boldsymbol{\xi}^*)$. Let us compute the Jacobian of the dynamical system equation 10. Denoting $\boldsymbol{A} = \partial a(\boldsymbol{\eta}^*)$ $\boldsymbol{B} = \partial b(\boldsymbol{\xi}^*)$ we have the following Jacobian

$$
\boldsymbol{C} = \begin{pmatrix} 1-\alpha & \alpha\boldsymbol{A} \\ (1-\alpha)\beta\boldsymbol{B} & (1-\beta)+\alpha\beta\boldsymbol{B}\boldsymbol{A} \end{pmatrix}
$$

Due to the Lipschitz constraint on the function $a$ and $b$, we have $\|\boldsymbol{A}\| \le L_a$ and $\|\boldsymbol{B}\| \le L_b$. According to Lemma 4, this leads to:

$$
\rho(\boldsymbol{C}) \le \rho\left( \begin{pmatrix} 1-\alpha & \alpha\|\boldsymbol{A}\| \\ \|(1-\alpha)\beta\boldsymbol{B}\| & \|(1-\beta)+\alpha\beta\boldsymbol{B}\boldsymbol{A}\| \end{pmatrix} \right) \le
$$
$$
\rho\left( \begin{pmatrix} 1-\alpha & \alpha L_a \\ (1-\alpha)\beta L_b & (1-\beta)+\alpha\beta L_b L_a \end{pmatrix} \right),
$$

and finally to $\rho(\boldsymbol{C}) < 1$, local asymptotic stability around the fixed point, as well as global asymptotic stability according to Lemmas 2 and 3. □

## C  ON THE FORM OF THE VARIATIONAL POSTERIOR

**Post-training conditional inference** A broad range of variational algorithms impose a fully factorisable form for the approximate posterior, which is usually referred to as the mean-field assumption (Hoffman et al., 2013; Blei et al., 2017; Kingma & Welling, 2013; Jaakkola & Jordan, 1996; Wainwright & Jordan, 2008; Knowles & Minka, 2011). In this context, one would usually opt in favour of a diagonal Gaussian distribution for $q_\theta$ base distribution. To relax this assumption and bring flexibility to the variational posterior, we relied on Householder flows (Tomczak & Welling, 2016) as a sparse approximation to a full covariance matrix:

$$\boldsymbol{x} = \boldsymbol{\mu} + \prod_{i=1}^{n} \boldsymbol{H}_i D(\boldsymbol{\sigma}) \boldsymbol{\epsilon} \tag{13}$$

where the function $D$ maps the vector $\boldsymbol{\sigma}$ to a diagonal matrix. Householder flows consist of a series of $n$ orthogonal transformation $\boldsymbol{H}_i$ of a base vector $\boldsymbol{\epsilon} \sim \mathcal{N}(0, \boldsymbol{I})$. In principle, such maps can model any orthogonal matrix when $n$ is sufficiently large. The main advantage of this approximate posterior formulation is its time and memory computational cost, as it only requires inner vector-vector and vector-scalar products.

**Learning from incomplete data** A common usage in VAEs is to amortise the construction of parametric variational posteriors by building a so-called inference network that maps each datapoint to its correspondent parameter configuration. In the case of incomplete data, one usually has to deal with missing values that are randomly spread in the data space and whose dimensionality can be hard to foreseen. Inputting such sparse data in a neural network can be challenging. To solve this, we filled every missing value with the median value of the data tensor, thereby creating a workable input for the inference network. In turn, this inference network produced a fully-factorised variational posterior which was indexed according to the latent space partition.

## D  MNIST DATA COMPLETION WITH IMPLICIT NORMALIZING FLOWS

We tested the VISCOS Flow algorithm on a data completion task with Implicit Normalizing Flows (INF) (Lu et al., 2021) to show that our method can be extended beyond iResNet architectures. Unlike iResNet, the Lipschitz constant of a single block of INF is unbounded, making it potentially considerably more expressive. To efficiently train INF, we note that the original problem that is to be solved for one block, which reads

$$F(\boldsymbol{x}, \boldsymbol{z}) = \boldsymbol{x} - \boldsymbol{z} + f(\boldsymbol{x}) - g(\boldsymbol{z}) = 0$$

is equivalent to solving

$$\boldsymbol{z} = h^{-1}(\boldsymbol{x} + f(\boldsymbol{x}))$$

where $h(\boldsymbol{z}) = \boldsymbol{z} + g(\boldsymbol{z})$ is a iResNet block. We know how to invert $h$ (Behrmann et al., 2019) using fixed-point iterations, and we have already presented how to propagate gradients through this transformation using Neumann series (see Section 3). These two solutions showed to be considerably faster and more memory efficient than the joint use of Brodyen method and linear system solving algorithms presented in the original work.

We trained a convolutional INF network that was defined in a similar manner as the iResNet model designed for the CIFAR task in the main text, with the following notable amendments. First, to match the INF structure, the direction of one every two iResNet blocks was flipped ($f_i \rightarrow f_i^{-1}$) such that plain and inverted residual blocks alternated. ActNorm layers (Kingma & Dhariwal, 2018) were interleaved in between iResNet blocks. To compute the higher order gradients required in Equation 8, and since the gradient of inverted iResNet block was computed using Neumann series, we relied once more on the following identity:

$$\nabla_{\boldsymbol{y}} \boldsymbol{G}(\boldsymbol{y}) = \nabla_{\boldsymbol{y}} \left( -\boldsymbol{G} \, \boldsymbol{J}(\boldsymbol{x}(\boldsymbol{y})) \, \boldsymbol{G} \right)$$

where $\boldsymbol{G} = \boldsymbol{J}^{-1} = (\boldsymbol{I} + \nabla_{\boldsymbol{x}} f(\boldsymbol{x}))^{-1}$ is the inverse Jacobian obtained through the truncated series $\sum_{j=0}^{n} -(\nabla_{\boldsymbol{x}} f(\boldsymbol{x}))^j$, and the dependence of $\boldsymbol{G}$ (or $\boldsymbol{x}$) on $\boldsymbol{y}$ is made explicit where needed. We trained this INF network on the complete MNIST training dataset using the Adam optimiser for 20 epochs with a learning rate of $10^{-3}$. We used the VISCOS Flow algorithm to complete digits

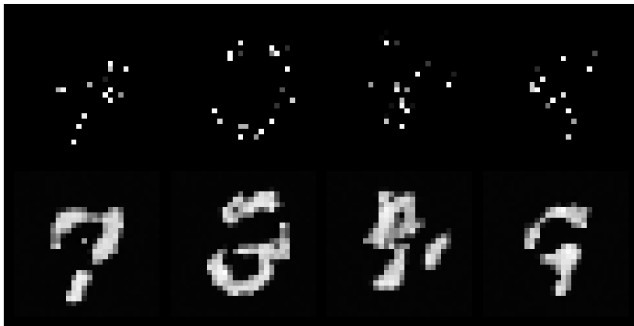

Figure 5: Implicit Normalizing Flow data completion results. Incomplete and completed digits are shown on the first and second row, respectively.

where 90% of the pixels were missing. The CLADE gradient estimator was used to approximate the gradient of the LAD of the partial Jacobian. Results of this task are displayed in Figure 5. We noticed that the fixed-point algorithm was slightly less effective with INF networks than with iResNet: for a small proportion of the iterations during the optimisation ($< 30\%$), our algorithm did not converge to the solution and required a few Newton-Krylov iterations ($< 5$) to complete the process. It is worth emphasising that, in all cases, the fixed-point algorithm always played a major role in finding the solution of the equality constraint, reducing the distance between the imposed values $\boldsymbol{y}^O$, $\boldsymbol{x}^H$ and their estimate $f^O(\widetilde{\boldsymbol{x}})$, $g^H(\widetilde{\boldsymbol{y}})$ by more than 90% with respect to the original values. Also, using the same partition of the observed and latent space led us to find a workable solution in every single completion case.

