# OpenReview forum: "VISCOS Flows: Variational Schur Conditional Sampling with Normalizing Flows"
_ICLR.cc/2022/Conference — ICLR 2022 Submitted_

### Official Review · Reviewer_puL5 · 2021-11-01

**Correctness:** 4
**Technical Novelty And Significance:** 3
**Empirical Novelty And Significance:** Not applicable
**Recommendation:** 3
**Confidence:** 3

**Main Review:**

### Strengths:
- AFAIK this is a novel approach to parameterising a variational distribution for inference within a normalising flow.
- I think that parameterising variational distributions for normalizing flows in this way is an interesting avenue to explore, and the derivation of this method and its gradient estimators is clearly non-trivial.

### Weaknesses:
- More comparisons to (or at least discussion of) related work should be included. [1] also perform imputation using variational inference in normalizing flows (although with a relaxed likelihood), and present (at least visually) better results than this paper. [2] present a conditional VAE that can be trained with incomplete training data.
- This approach seems to be fundamentally difficult to scale to higher-dimensional datasets (due to the cost of solving the equality constraint and estimating the gradient). Even for the moderately high-dimensional CIFAR10 and MNIST datasets experimented on, I suspect that alternative techniques (like [1,2]) work better. A much better case needs to be made for why this method should be preferred.

### Minor
- last 3 lines of first paragraph of Section 2.4 - This sentence is confusing. An ELBO of zero is not meaningful for continuous data, since the likelihood may be above or below zero. And it is not clear what is meant by "whenever $x^O$ is independent of $x^H$": should they be independent under the prior, or under the posterior conditioned on observations?
- I think $\widetilde{\mathbf{y}^H}$ is used before being defined in Algorithm 1: it is used on the RHS of line 8 but has no assigned value until after line 8 is executed.

[1] Whang, Jay, Erik Lindgren, and Alex Dimakis. "Composing Normalizing Flows for Inverse Problems." International Conference on Machine Learning. PMLR, 2021.

[2] Ma, Chao, et al. "Eddi: Efficient dynamic discovery of high-value information with partial vae." arXiv preprint arXiv:1809.11142 (2018).

**Summary Of The Paper:**

This paper proposes a technique for performing data imputation using pre-trained normalizing flows. They do so by fitting a variational distribution over a subset of the 'base' variables of the normalizing flow. Along with the observations, a sample from this variational distribution is sufficient to fully specify a sample from the normalizing flow model. The authors present techniques for constructing samples in this way, and computing the derivatives required to optimise the variational distribution.

**Summary Of The Review:**

While I appreciate the novel idea  and derivations, it is not clear to me that there are any use cases where this method should be preferred over the alternatives. Given the lack of a convincing case for its usefulness, I am recommending reject.

---

> ### Author Response · Authors · 2021-11-22
> **Response**
>
> We want to thank the reviewer for a careful evaluation of the manuscript and constructive feedback. We opted for including only short answers to reviewers. We made some minor changes (in magenta) in the pdf file.
>
> We want to apologize for the confusion created by re-editing. Our approach can work with both pre-trained normalizing flows and added to training of the normalizing flow (i.e., the ELBO is simply subtracted from the training loss). In fact, in one of our experiments we trained the flow together with the conditional sampler.
>
> We have corrected the minor problems for readability of the manuscript.
>
> Regarding the rest of the comments. Unfortunately, we cannot perform additional experiments at this time. We would like to point out the main difference to [1] is that our approach offers further flexibility as the observational masks can be changed after training, while in [1] they have to be specified, as far as we can understand. The work in [2] seems to be closely related to ACFLOW and we have discussed the differences to the work in the introduction. In general, VISCOS flows can address several problems e.g.,: sampling with pre-trained NF, training with incomplete data and in-painting, while ACFLOW are trained for a specific task. Hence, our approach offers a versatile model that can perhaps show lower performance on specific tasks, but it can be applied to a number of different tasks.
>
> [1] Whang, Jay, Erik Lindgren, and Alex Dimakis. "Composing Normalizing Flows for Inverse Problems." International Conference on Machine Learning. PMLR, 2021.
>
> [2] Ma, Chao, et al. "Eddi: Efficient dynamic discovery of high-value information with partial vae." arXiv preprint arXiv:1809.11142 (2018).

---

> > ### Comment · Reviewer_puL5 · 2021-11-26
> > **Rebuttal reply**
> >
> > Thank you to the authors for their response. Having considered it, I appreciate that there is value in not requiring the mask distribution to be specified at training time. However I don't think that this value is well-enough demonstrated in the experiments on image completion given the simple masks used (given that baselines like [1,2] could be trained with complex mask distributions such as in [3]), and the fairly weak empirical results. I have therefore not changed my recommendation.
> >
> > [1] Whang, Jay, Erik Lindgren, and Alex Dimakis. "Composing Normalizing Flows for Inverse Problems." International Conference on Machine Learning. PMLR, 2021.
> >
> > [2] Ma, Chao, et al. "Eddi: Efficient dynamic discovery of high-value information with partial vae." arXiv preprint arXiv:1809.11142 (2018).
> >
> > [3] Zhao, Shengyu, et al. "Large scale image completion via co-modulated generative adversarial networks." arXiv preprint arXiv:2103.10428 (2021).

---

### Official Review · Reviewer_urKg · 2021-11-02

**Correctness:** 4
**Technical Novelty And Significance:** 2
**Empirical Novelty And Significance:** 1
**Recommendation:** 3
**Confidence:** 4

**Main Review:**

The paper is relatively well written although I believe that parts of writing in the paper can be made more precise that will help the overall quality of the paper. The problem formulation is clear and the proposed solution is interesting as well as explored in detail.

However, the paper suffers from several weaknesses:

1) Pre-trained flows: The paper does not train a normalizing flow for data that may itself have missing features/data but assumes access to a pre-trained flow and uses it for sampling when some features are missing. I find this set-up a bit unnatural since I'd expect that for application where such a conditional sampling might be required, the training data for learning any probability model will itself have missing features.

2) Restricted flow architectures: Secondly, the proposed solution works for only a restricted class of flow architectures ie those that have residual flow (or iResNet) like structures. I think this is a major limitation of the method and there is no clear discussion or solution of how the proposed method can be extended for a general class of flow models.

3) Comparison to ACFlow and MCFlow: The authors in the introduction and as part of the motivation highlight why these models are not well suited for the problem they consider. They mainly highlighted computational cost etc as a major roadblock to deploying such models. However, unfortunately, this claim has not been substantiated with any justification. I believe clear empirical analysis is required to include such a claim and to highlight why VISCOS flows a re a better alrenative to the already proposed models in the literature.

4) Experiments: I also believe that the present experiments are insufficient to evaluate the performance of VISCOS flows. Particularly, condiering the restrictive nature of the flow architectures that VISCOS flows use, I think the present experiments do not fully reveal the advantages of different methods. Precisely, since the experimental results obtained for methods like PL-MCMC, MFlow use a different flow architecture than for the results for VISCOS flows, I believe the experimental comparisons are not fair and sufficient enough to draw any meaningful conclusions. Furthermore, the results themselves are mostly mixed and due to the lack of any error bars for these results it is hard to conclude that nay method may be better than the others. I'd recommend the authors to atleast perform experiments that uses the same flow architecture to make the comparisons fair and if possible include error bars for the results.

**Summary Of The Paper:**

The paper proposes a method to sample from conditional distributions using a pre-trained flow based method for applications in problems that have missing features. The proposed method utilizes a lower bound on the log-probability of the conditional distribution (conditioned on observed features) using the Schur complement.

**Summary Of The Review:**

The paper studies an interesting problem for flows and proposes a nice solution. However, in the current form, the paper suffers from many weaknesses surrounding the experimental setup and being valid for only a restrictive class of flow models.

---

> ### Author Response · Authors · 2021-11-22
> **Response**
>
> We want to thank the reviewer for a careful evaluation of the manuscript. We opted for including only short answers to reviewers. We made some minor changes (in magenta) in the pdf file.
>
> We want to apologize for the confusion created by re-editing. Our approach can work with both pre-trained normalizing flows and added to training of the normalizing flow (i.e., the ELBO is simply subtracted from the training loss). In fact, in one of our experiments we trained the flow together with conditional sampling.
>
> It is true that we have not presented results for other architectures (such as GLOW), however, our approach is more flexible than others in this sense. Indeed, our ELBO loss can be added to other normalizing flows during training, while MCLFOW and ACFLOW are restricted to a specific architectures.
>
> Regarding the rest of the comments. Unfortunately, we cannot perform additional experiments at this time.

---

### Official Review · Reviewer_7gsd · 2021-11-03

**Correctness:** 3
**Technical Novelty And Significance:** 3
**Empirical Novelty And Significance:** 1
**Recommendation:** 5
**Confidence:** 3

**Details Of Ethics Concerns:**

This is not applicable to the ethical considerations

**Main Review:**

1.
This paper provides an interesting utilization of the Schur complement to model the dimensions of observed and unobserved features.
I cannot understand the notation of $G^{HH}(x)$ in Eq 2 because $G$ is defined on $y$.

2.
This paper derives the gradient of the suggested ELBO formulation. This derivation results in some theoretic suggestions, such as the corrected gradient estimators for the log-absolute determinants, i.e. CLADE.

3.
It would have been better to show an illustrative figure on the suggested Flow structure. Particularly, the structure is decomposed into the observed and the unobserved variables; and this decomposition leads to the link to the Schur decomposition. It is confusing which Jacobian sub-matrix is related to which variable.

4.
The in-painting task is compared from the perspective of classification accuracy, FID, and RMSE. There are some other performance standards, i.e. SSIM, PSNR. I wonder whether these metrics could be used or not.[1]

5.
Currently, only MNIST is being used for the test. Previous works utilize more diverse datasets, such as places2, celeba, streetview, etc.[2]

6.
Benchmark baseline models are too few. if you search inpainting flow models, there will be plenty of recent flow models for the same task.


[1] Xu, Rui, et al. "Deep flow-guided video inpainting." Proceedings of the IEEE/CVF Conference on Computer Vision and Pattern Recognition. 2019.
[2] Ren, Yurui, et al. "Structureflow: Image inpainting via structure-aware appearance flow." Proceedings of the IEEE/CVF International Conference on Computer Vision. 2019.





**Summary Of The Paper:**

This paper presents a variant of flow, VISCOS flow, utilizing the Schur complement. This flow model is applied to the in-painting task, which estimates the unobserved feature variables given the observed features. Given this in-painting task, the inference requires the modeling on the latent variable, which results in the ELBO derivation on such latent variables as variational variables.

**Summary Of The Review:**

This paper provides an interesting utilization of Schur complement in the in-painting task. However, the evaluation is very weak.

////////////////////

I read the rebuttal from the authors, and I maintain my evaluation.

---

> ### Author Response · Authors · 2021-11-22
> **Response**
>
> We want to thank the reviewer for a careful evaluation of the manuscript. We opted for including only short answers to reviewers. We made some minor changes (in magenta) in the pdf file.
>
> We adjusted the equation 2, by changing $G^{HH}(x)$ to $G^{HH}(f(x))$, apologies for confusion!
>
> We appreciate that the notation is quite heavy and can be very confusion. However, we have not found a better way to present our results. Regarding an illustrative figure, we thought that Figure 1 can serve this purpose.
>
> Regarding the rest of the comments. Unfortunately, we cannot perform additional experiments at this time.

---

> > ### Comment · Reviewer_7gsd · 2021-11-25
> > **Thanks**
> >
> > Thanks for the response, and I maintain my evaluation because of the weak empirical results

---

### Decision · Program_Chairs · 2022-01-20

**Decision:**

Reject

**Comment:**

This paper proposes a technique to perform data imputation with normalizing flow defining a joint density between observed and unobserved variables. This is achieved by introducing a variational posterior over the missing variables which is parametrized in terms of the original model by using the Schur complement of the model's Jacobian over the hidden variables.
The idea is interesting, but the proposed setup is quite complex and the experimental results are not conclusive. The quality of the results shown can likely be matched or surpassed with much simpler techniques. The paper would substantially benefit from more detailed experiments.